# Autotrophic growth of *Escherichia coli* is achieved by a small number of genetic changes

**Roee Ben Nissan[1†], Eliya Milshtein[1†], Vanessa Pahl[2], Benoit de Pins[1], Ghil Jona[3], Dikla Levi[3], Hadas Yung[1], Noga Nir[1], Dolev Ezra[1], Shmuel Gleizer[1], Hannes Link[2], Elad Noor[1], Ron Milo[1]***

[1]Department of Plant and Environmental Sciences, Weizmann Institute of Science, Rehovot, Israel; [2]Interfaculty Institute for Microbiology and Infection Medicine Tübingen, University of Tübingen, Tübingen, Germany; [3]Department of Life Sciences Core Facilities, Weizmann Institute of Science, Rehovot, Israel

**Abstract** Synthetic autotrophy is a promising avenue to sustainable bioproduction from $CO_2$. Here, we use iterative laboratory evolution to generate several distinct autotrophic strains. Utilising this genetic diversity, we identify that just three mutations are sufficient for *Escherichia coli* to grow autotrophically, when introduced alongside non-native energy (formate dehydrogenase) and carbon-fixing (RuBisCO, phosphoribulokinase, carbonic anhydrase) modules. The mutated genes are involved in glycolysis (*pgi*), central-carbon regulation (*crp*), and RNA transcription (*rpoB*). The *pgi* mutation reduces the enzyme's activity, thereby stabilising the carbon-fixing cycle by capping a major branching flux. For the other two mutations, we observe down-regulation of several metabolic pathways and increased expression of native genes associated with the carbon-fixing module (*rpiB*) and the energy module (*fdoGH*), as well as an increased ratio of $NADH/NAD^+$ - the cycle's electron-donor. This study demonstrates the malleability of metabolism and its capacity to switch trophic modes using only a small number of genetic changes and could facilitate transforming other heterotrophic organisms into autotrophs.

## eLife assessment

This is an **important** follow-up study to a previous paper in which the authors reconstituted $CO_2$ metabolism (autotrophy) in *Escherichia coli*. Here, the authors define a set of just three mutations that promote autotrophy, highlighting the malleability of *E. coli* metabolism. The authors make a **convincing** case that mutations in pgi are loss-of-function mutations that prevent metabolic efflux from the reductive pentose phosphate autocatalytic cycle, and their data suggest possible roles of mutations in two other genes - *crp* and *rpoB*. This research will be particularly interesting to synthetic biologists, systems biologists, and metabolic engineers aiming to develop synthetic autotrophic microorganisms.

## Introduction

Anthropogenic interference and intensive burning of fossil fuels releases large amounts of sequestered carbon in the form of $CO_2$, a main driver of climate change (*Olofsson and Hickler, 2008*; *Friedlingstein et al., 2022*). Direct air carbon capture, by any means, is urgently needed to help reduce emissions. Here, we focus on biology-based carbon fixation as a potential avenue for $CO_2$ capture, which has been shown to be a potentially efficient production platform (*Leger et al., 2021*;

*For correspondence:
ron.milo@weizmann.ac.il

†These authors contributed equally to this work

*Van Peteghem et al., 2022*; *Schwander et al., 2016*). Specifically, we suggest using microbes for $CO_2$ valorization and produce a diverse repertoire of products including food, fuel, bio-plastics or other commodities (*Claassens et al., 2019*; *Gleizer et al., 2020*; *Li et al., 2012*; *Nielsen and Keasling, 2016*). Natural autotrophs already metabolise $CO_2$ efficiently, yet most are challenging to cultivate and manipulate genetically compared to heterotrophic model organisms like *Escherichia coli*. Here, we explore an emerging alternative approach – converting heterotrophic model organisms into tractable autotrophs for bio-production (*Gleizer et al., 2020*; *Baumschabl et al., 2022*; *Gassler et al., 2020*; *Gleizer et al., 2019*). In addition to promising engineering applications, studying the transition from heterotrophic to autotrophic metabolism can uncover fundamental principles about the structure and regulation of carbon fixation metabolism (*Gleizer et al., 2019*; *Gassler et al., 2020*; *Antonovsky et al., 2016*; *Barenholz et al., 2017*; *Flamholz et al., 2020*; *Herz et al., 2017*).

Recently, with the increasing interest in platforms for sustainable production, synthetic biologists have been able to successfully manipulate different organisms to enable major metabolic transitions (*Gassler et al., 2020*; *Gleizer et al., 2019*; *Kim et al., 2020*; *Chen et al., 2020*; *Keller et al., 2022*; *Bang et al., 2020*; *Satanowski et al., 2020*; *Yishai et al., 2018*). These efforts resulted in the successful introduction of non-native C1 utilisation pathways, and even included successful transitions from heterotrophic into completely autotrophic metabolisms in bacteria (*Gleizer et al., 2019*) and yeast (*Gassler et al., 2020*). These previous studies introduced the reductive pentose phosphate (rPP) cycle, the most prominent carbon fixation pathway in nature (also known as the Calvin-Benson cycle), into *Escherichia coli* (*E. coli*) and *Komagataella phaffii* (previously *Pichia pastoris*). Indeed, we showed that *E. coli* was able to utilise the rPP cycle for the synthesis of all biomass carbon - thus converting it from a heterotroph to an autotroph (*Gleizer et al., 2019*). However, this transition required continuous culture in selective conditions for several months (adaptive lab evolution) and resulted in many uninterpretable mutations, exposing a large knowledge gap.

Using adaptive lab evolution presents a challenge as it is hard to isolate the changes necessary for a given phenotype from the overall set of accumulated mutations. Here, we aimed to characterise the landscape of mutations required for autotrophic growth. Studying mutations that arise in multiple lineages can help us better understand the biological basis for the heterotroph-to-autotroph transition. To achieve this goal, we created a pipeline that enabled us to distinguish mutations essential for autotrophy from unspecific low-frequency mutations, that is ones that are generally promoted by the physiological conditions in evolution experiments. The final outcome of this pipeline is a compact essential set of changes permitting autotrophy in *E. coli*. We then proceeded with further analyses to shed light on the mechanism behind this compact set of essential mutations. We show the phenotype included a change in the activity of a central carbon metabolism enzyme and intracellular changes of the non-native carbon fixation cycle's metabolite and co-factors pools. We conclude by speculating about the role of these metabolic changes.

## Results

### A compact set of mutations that enable autotrophic growth was identified using rational design, iterative lab evolution and genetic engineering

Adaptive lab evolution is an effective tool that enables integration of non-native metabolic pathways. In order to harness the power of adaptive lab evolution, a selection system must be implemented to direct the evolution towards the desired function. In our case, the selection for the activity of the non-native rPP cycle. To ensure that some level of carbon fixation by RuBisCO becomes essential for growth, we knocked out phosphofructokinase (*pfkA* and *pfkB*), and glucose-6-phosphate dehydrogenase (*zwf*) which creates a stoichiometric imbalance and growth arrest when growing on five-carbon sugars such as xylose (*Gleizer et al., 2019*). This imbalance could be rescued by a metabolic bypass - the phosphorylation of ribulose-5-P by Prk and the subsequent carboxylation by RuBisCO, thereby coupling growth to the heterologously expressed rPP enzymes. Critically, this setup allowed us to evolve *E. coli*, a natural heterotroph, into an autotroph, using a non-native rPP cycle to utilise $CO_2$ as a sole carbon-source. This system also requires formate dehydrogenase to oxidise formate for energy (*Gleizer et al., 2019*).

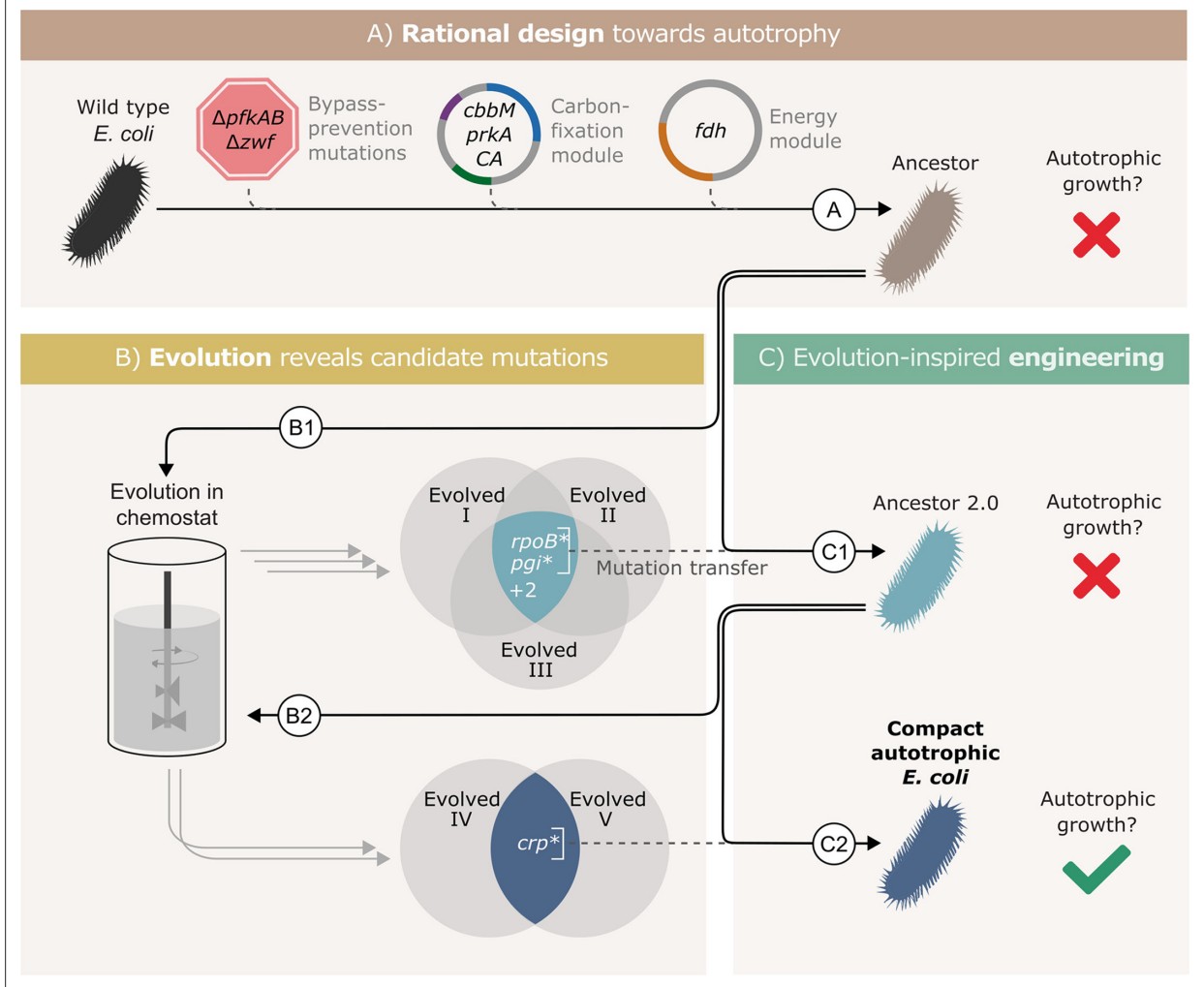

**Figure 1.** Autotrophic phenotype achieved by introducing 3 mutations on top of a rationally designed ancestor. (**A**) We rationally designed the 'wild-type' *E. coli* background strain (BW25113, depicted as a black bacterium) by introducing 4 enzymes: RuBisCO (*cbbM*), phosphoribulokinase (*prkA*), carbonic anhydrase (*CA*) and formate dehydrogenase (*fdh*), and 3 genomic knockouts: glucose-6-phosphate dehydrogenase (*zwf*) and phosphofructokinase A&B (*pfkAB*). We denote the resulting strain as 'Ancestor' (brown bacterium). (**B**) We tested the resulting strain for autotrophic growth and, since it didn't grow, we used it for adaptive lab evolution in xylose-limited chemostats - that is, conditions that select for higher carbon fixation flux. Altogether, we were able to isolate three evolved clones (two distinct strains from one chemostat experiment, Evolved I and II, and another strain from a second experiment, Evolved III, step B1). (**C**) Out of four consensus mutations that we identified, two - *rpoB* and *pgi* - were incorporated into the ancestor, giving 'Ancestor 2.0' (light blue bacterium, step C1). Once more, we tested for autotrophic growth and were unsuccessful. Therefore, we initiated another round of adaptive lab evolution experiments using Ancestor 2.0 growing in two xylose limited chemostats and isolated two evolved clones (one from each chemostat, step B2). The two clones shared a single consensus mutation, in *crp*. We thus created a new strain from Ancestor 2.0 by introducing the *crp* mutation into it (blue bacterium, step C2). This strain could grow autotrophically and thus we achieved a compact autotrophic strain.

Although adaptive lab evolution is a powerful tool, it also presents a challenge - once a new phenotype has been achieved, it is hard to separate the essential changes required for its appearance from the overall set of accumulated mutations. In many cases, mutations occur on the background of highly adaptive ones, leading to a spread of neutral 'hitchhiker' mutations in the population. In other cases, mutations that appeared at the beginning of the evolution are no longer needed for the final phenotype. Our goal was to find a set of mutations that, if introduced into a heterotrophic wild type *E. coli*, would result in the successful integration of the non-native carbon fixation and energy modules. This would enable autotrophic growth based solely on genetic engineering without the need for further evolutionary adaptation and potentially uncovering the mechanism behind this metabolic transition.

To isolate the essential genetic changes, we used a workflow that included three main stages as shown in *Figure 1*. (A) rational design - introduction of required heterologous genes and metabolic knockouts to enforce RuBisCO-dependent growth; (B) adaptive lab evolution -revealing mutation candidates by growing the designed strain in autotrophic-selecting conditions over many generations until an autotrophic phenotype is obtained; (C) evolution-inspired genetic engineering - introducing the most promising mutations revealed in stage B into the designed strain and testing it for autotrophic growth. Stages B and C were repeated until no further evolution was required, that is the designed strain in stage C had an autotrophic phenotype.

To find the most promising mutations in stage B, we applied at least one of two criteria: mutations in genes that occur repeatedly in different autotrophic lineages (but are uncommon in other adaptive lab evolution experiments) or mutations that when reverted, result in a loss of phenotype. We discovered these 'consensus' mutations by sequencing isolated autotrophic clones, obtained from repeated adaptive lab evolution experiments - similar to the conditions we used in *Gleizer et al., 2019*; *Figure 1*; steps B1, B2.

In practice, we inoculated the strain denoted as 'Ancestor' into a chemostat with limited xylose and excess formate. After ≈3 months (≈60 chemostat generations) we isolated from the chemostat a clone able to grow under autotrophic conditions (methods), and compared its mutation set to those of two previously evolved autotrophic clones (*Gleizer et al., 2019*). The intersection of the three clones contained four genes that were mutated in all autotrophic isolated clones: the central carbon metabolism enzyme phosphoglucoisomerase (*pgi*), the beta subunit of the RNA polymerase (*rpoB*), poly A polymerase (*pcnB*) and the MalT DNA-binding transcriptional activator (*malT*) (*Figure 1*, step B1). *rpoB*, *pcnB* and *malT* are genes that are commonly mutated in adaptive lab evolution experiments (*Phaneuf et al., 2019*), suggesting that they have little relevance for autotrophic growth. Therefore, we chose to focus on the *pgi* mutation.

We introduced the H386Y *pgi* mutation into the rationally designed ancestor (*Figure 1*, step C1), and the engineered strain was tested for growth in autotrophic conditions (Materials and methods). No growth was observed in those conditions, which meant that additional mutations were necessary in order to achieve autotrophy.

Genomic sequencing revealed that during the genetic manipulation for introducing the *pgi* mutation, another mutation in *rpoB* (A1245V) appeared in the genome. In parallel, reverting a variety of RNA polymerase mutations in other evolved strains back to their wild-type allele, showed that mutations in RNA polymerase are in fact essential to the phenotype. Therefore, despite the fact that it was unintentional, and different from the other *rpoB* mutations observed in the ALE experiments, we left the mutation in the genome and proceeded with this strain.

This strain, mutated in both *pgi* and *rpoB* (denoted 'Ancestor 2.0') was then used as the starting point for another round of evolution (*Figure 1*, step B2). Since it had already contained two mutations that were likely to be relevant for the desired phenotype, we expected to observe evolved autotrophic strains after fewer generations, compared to the first round, and to find fewer mutations in them. Indeed, on two separate experiments, it took roughly 1 month until autotrophic growth was observed, as opposed to more than 90 days in previous attempts. Even though these strains (*Figure 1*, 'Evolved 4' and 'Evolved 5') both had multiple mutations, only one mutation was shared between these independent evolution experiments - a non-synonymous mutation in the cAMP receptor protein, *crp* (H22N). When we introduced this mutation to the 'Ancestor 2.0' background in the evolution-inspired genetic engineering stage (*Figure 1*, step C2), the strain was immediately able to grow under autotrophic conditions.

Since the RNA polymerase mutation was introduced into this strain (*rpoB* A1245V) as a byproduct of genetic engineering and never observed in the adaptive evolution experiments, we needed to verify its essentiality to the autotrophic phenotype. We found that a reversion of *rpoB* A1245V back to the wild-type allele causes a loss of the autotrophic phenotype. We therefore could conclude that three mutations (*pgi**, *crp**, and *rpoB**) are sufficient to facilitate the autotrophic phenotype on top of the heterotrophic ancestor background. We denote the final engineered autotrophic *E. coli* strain, the 'Compact autotrophic *E. coli*'.

We verified the genotype using whole genome sequencing (Materials and methods). The results included the rationally designed knockouts (Δ*pfkA*, Δ*pfkB*, Δ*zwf*), heterologous plasmids (energy and carbon-fixing modules) and the three introduced mutations (*pgi**, *crp**, and *rpoB**). During the genetic

engineering process, two additional mutations occurred unintentionally in the genes *uhpT* and *yejG*. *uhpT* is a hexose transporter and is unlikely to have any effect on the autotrophic phenotype, especially because the mutation was an early stop codon (Q7*) and likely a loss of function. As the function of *yejG* is unknown, we wanted to ensure that it is not essential to the autotrophic phenotype. Therefore, we reverted the mutation back to its wild-type allele and found that indeed the cells were still autotrophic.

Thus, we constructed a new autotrophic strain with three essential mutations (*pgi*, *crp*, and *rpoB*), which we denoted the *compact* strain. When comparing this compact strain to a previously characterised autotrophic strain (*Gleizer et al., 2019*), it was nearly identical in terms of growth rate, lag time, and yield (*Figure 2A*). Using $^{13}C$ labeling, we verified that all the biomass carbon stems from $CO_2$ (*Figure 2B*). We termed the three introduced mutations (*pgi*, *crp,* and *rpoB*) together with the carbon fixation machinery (*cbbM*, *prkA,* and *CA*) and the energy module (*fdh*) the 'autotrophy-enabling gene set'.

## Introduction of autotrophy-enabling gene set into wild-type background does not require bypass-prevention for growth

As described above, in order to select for autotrophic growth, the native *E. coli* was metabolically re-wired by three auxiliary genomic knockouts (Δ*pfkA*, Δ*pfkB*, Δ*zwf*). These knockouts created a dependency on carboxylation by RuBisCO for growth even when consuming pentose sugars which were supplemented during the chemostat evolution (see methods, *Antonovsky et al., 2016*; *Gleizer et al., 2019*). This dependency was expected to direct the evolution towards increased usage of the synthetic rPP cycle and, eventually, to autotrophic growth.

Once autotrophic growth was achieved, and the autotrophy-enabling gene set (*cbbM*, *prkA, CA, fdh*, *pgi H386Y*, *crp H22N*, and *rpoB A1245V*) was found, we wanted to test if these auxiliary knockouts were still required for the phenotype. We introduced the set of autotrophy-enabling genes into a wild-type (BW25113) *E. coli* background (Materials and methods, *Figure 3*). After doing so, we discovered an unintentional genomic mutation, in the ribosomal 16 S subunit (rrsA) that was introduced during the genomic manipulation process. The engineered *E. coli* strain was able to grow autotrophically, although its final OD was half and its growth rate was less than half of that of the rationally designed compact strain still harbouring the auxiliary knockouts (*Figure 3—figure supplement 1*). This indicates that the autotrophy-enabling gene set is sufficient to achieve the trophic mode change and that the knockouts are helpful but not essential for the final autotrophic growth. As discussed below, this demonstrates that, in some cases, metabolic rewiring could be viewed as a temporary '*metabolic scaffold*' which can be removed from the genotype once the desired phenotype is achieved.

We thus established a new engineered autotrophic *E. coli* strain containing a set of three essential and sufficient mutations, and showed that the bypass-prevention knockouts introduced before the ALE are themselves not necessary for the phenotype. Due to the small required number of genetic modifications, this strain provides an opportunity to derive guidelines for future engineering efforts that use the rPP cycle. Having this goal in mind, we focused our attention on identifying the relevant phenotypes of each of these mutations and explaining their adaptive advantage. Because the mutations were essential for the autotrophic growth, we could not use autotrophic conditions to compare them to a strain with wild-type alleles. Therefore, we chose the most suitable combinations of strains and conditions that could isolate the effect of the pertinent mutation.

## Modulation of a metabolic branch-point activity increased the concentration of rPP metabolites

Metabolic pathways that regenerate and synthesise more of their own metabolites are referred to as autocatalytic cycles. Within the autotrophic *E. coli* the rPP cycle is autocatalytic. Due to the inherent positive feedback mechanism, autocatalytic cycles tend to be unstable, and therefore the fluxes of entry and exit points (bifurcation/branch points) need to be balanced (*Barenholz et al., 2017*). Such tuning is needed since any disruption of the balance between cycling and branching fluxes could result in the depletion, or alternatively toxic accumulation of intermediate metabolites, and cycle arrest.

We previously predicted that mutations in branch points will be needed to stabilise the steady state flux within the rPP cycle (*Barenholz et al., 2017*). In line with this prediction we find that the mutation in phosphoglucoisomerase (Pgi), a key branch point in the rPP cycle, follows this design principle.

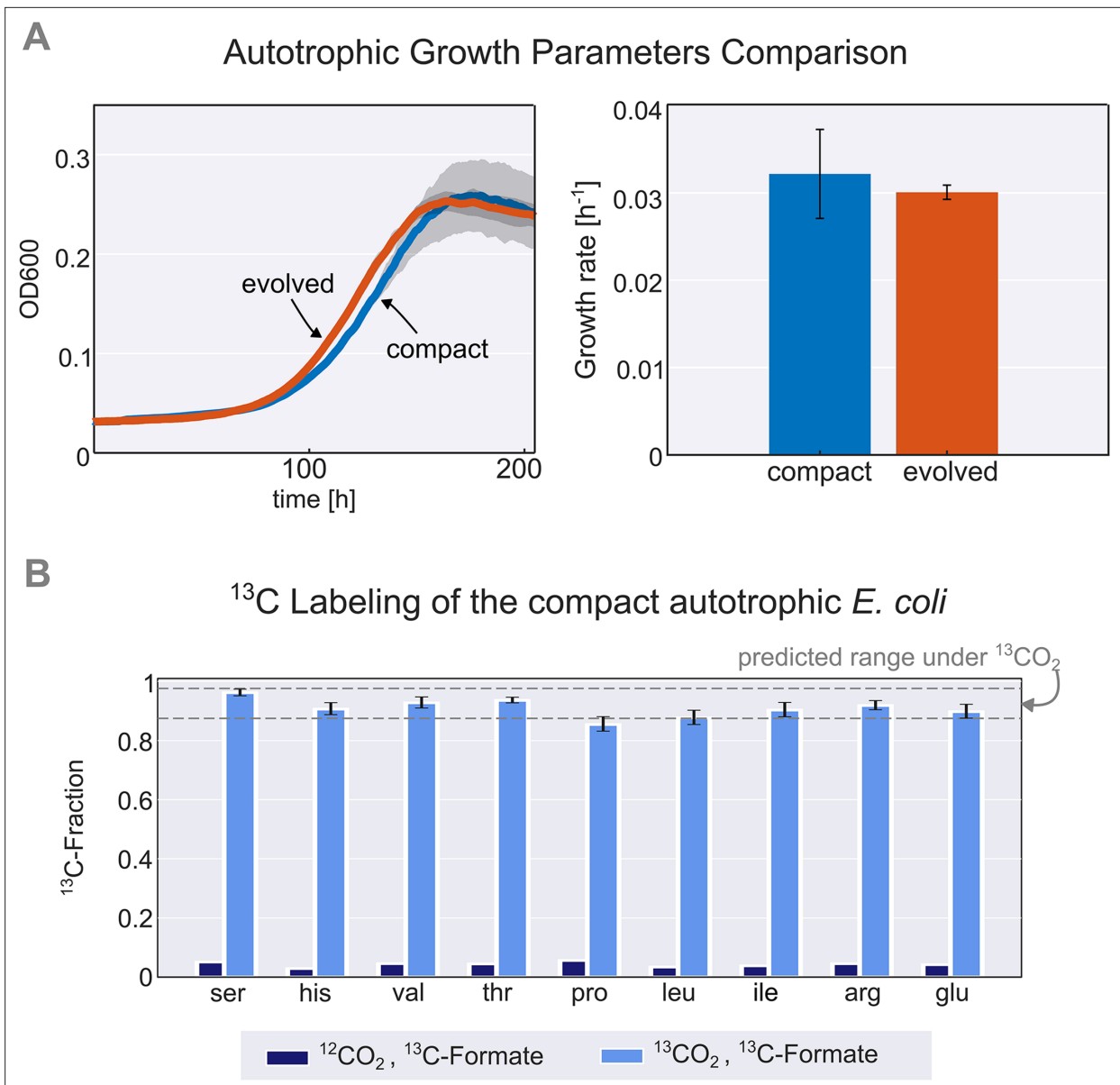

**Figure 2.** Validation and characterisation of the autotrophic phenotype. (**A**) Left: growth curve of an isolated evolved clone (orange) versus the engineered compact strain (light blue) in liquid M9 minimal media supplemented with 45 mM sodium formate and sparged with a gas mixture of 10% $CO_2$, 5% oxygen, and 85% nitrogen. Right: calculated growth rate using linear regression of the engineered compact strain and evolved strain, the calculated doubling time with the given conditions is about 24 hr ($\mu \approx 0.03$ h$^{-1}$). Growth was carried out in triplicates (n=3) in a Spark plate reader with gas control, dark grey area and error bars correspond to the standard deviation of the mean (150 µl culture +50 µl mineral oil, to prevent evaporation). (**B**) The fractional contribution of $^{13}CO_2$ to various protein-bound amino acids of the compact autotrophic strain after ≈4 doublings on $^{13}CO_2$ and $^{13}C$ labeled formate (light blue, mean of n=3). This reached the expected $^{13}C$ labeling fraction (see Materials and methods) of the biosynthesized amino acids. When grown in naturally labeled $CO_2$ and $^{13}C$ labeled formate (dark blue, n=1, technical triplicate) the $^{13}C$ content dropped to close to natural abundance. Experiments with $^{13}CO_2$ as the substrate were carried out in air-tight (i.e. sealed) growth vessels. Error bars represent standard deviation of the mean.

The online version of this article includes the following figure supplement(s) for figure 2:

**Figure supplement 1.** The fractional contribution of $^{13}CO_2$ to various protein-bound amino acids evolved autotrophic strain after ≈4 doublings on $^{13}CO_2$ and $^{13}C$ labeled formate (light blue, mean of n=3) reached the expected $^{13}C$ labeling fraction of the biosynthesized amino acids (see Materials and methods).

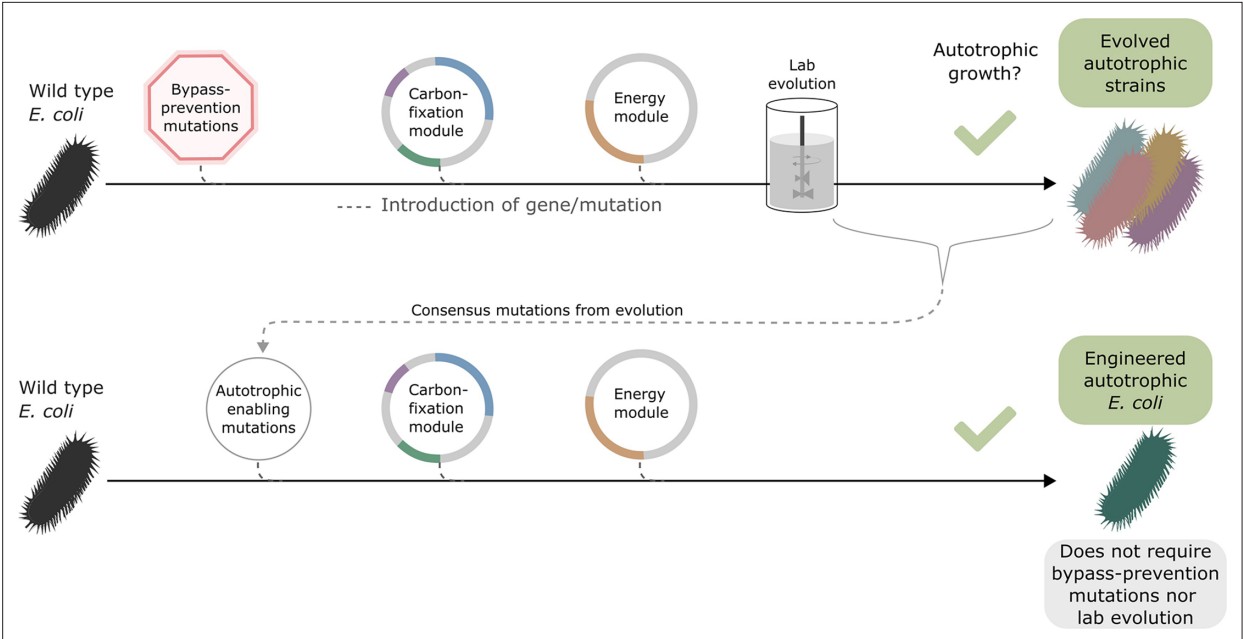

**Figure 3.** Auxiliary knockouts are not required for final phenotype. We transformed a wild-type BW25113 *E. coli* strain (black bacteria) with the carboxylating and energy module plasmids (grey circles with coloured gene annotations), and inserted three auxiliary genomic knockouts (red octagon) to rewire metabolism toward carboxylating dependency. This strain was used for iterative evolution experiments in order to generate diverse autotrophic strains and reveal mutations candidates for rational design, that is consensus mutations. The identified autotrophic enabling mutations (thin grey circle) were then introduced in a wild-type strain expressing the heterologous plasmids but without the auxiliary knockouts, the final strain was able to grow in autotrophic conditions. Dashed lines represent gene/mutation introduction.

The online version of this article includes the following figure supplement(s) for figure 3:

**Figure supplement 1.** Growth of Compact versus Engineered autotrophic *E. coli*.

Pgi consumes fructose-6-phosphate (F6P), one of the cycle intermediates, converting it into glucose-6-phosphate (G6P), a precursor for cell membrane biosynthesis. Thus, Pgi regulates flux out of the autocatalytic cycle (*Figure 4A*).

In the different adaptive lab evolution experiments, we found three distinct mutations in the *pgi* gene. The first, H386Y, is a non-synonymous mutation occurring in one of the catalytic residues in the active site, and is part of the autotrophy-enabling set. The second was a complete knockout, a 22 KB chromosomal deletion, including also 16 other genes. The third was an early stop codon E72*. These observations led us to suspect the H386Y mutation in *pgi* decreases or even completely eliminates the activity of the enzyme.

We tested the kinetic rates of the isomerization of F6P to G6P of two purified Pgi enzymes, wild type Pgi and Pgi H386Y, using a spectroscopic coupled assay, where G6P was coupled to $NADP^+$ reduction (*Widjaja et al., 1998*; *Figure 4B*; see also methods). The measured $k_{cat}$ of the wild type Pgi was $\approx 130$ [$s^{-1}$] ($\pm 26$) which is in line with previous computational predictions of a $k_{cat}$ of $\approx 200$ [$s^{-1}$] (*Davidi et al., 2016*). The mutated Pgi showed weak activity with a $k_{cat}$ of $\approx 1.5$ [$s^{-1}$]($\pm 1$). Therefore, the *pgi* mutation is in line with the original expectation that branching point regulation is necessary, and the nature of the observed mutations in *pgi* supports the notion that their role is to reduce the efflux from the non-native autocatalytic cycle.

Pgi is part of glycolysis/gluconeogenesis, where significant flux is required in wild type *E. coli*. Accordingly, wild type Pgi catalyses a reaction that is causing a strong efflux of F6P from the cycle, diminishing the regeneration of ribulose-1,5-bisphosphate (RuBP) which is usually not needed in the native metabolism but is essential for the rPP autocatalytic activity. In the mutated version of Pgi the efflux capacity is diminished, which can stabilise the cycle (*Figure 4C*).

Following the same logic, we can expect an increase in the F6P substrate metabolite pool due to reduced consumption by Pgi. Therefore, we measured intracellular sugar-phosphates in the *pgi* mutant strain and a strain with a wild-type *pgi* allele by liquid chromatography-tandem mass spectrometry

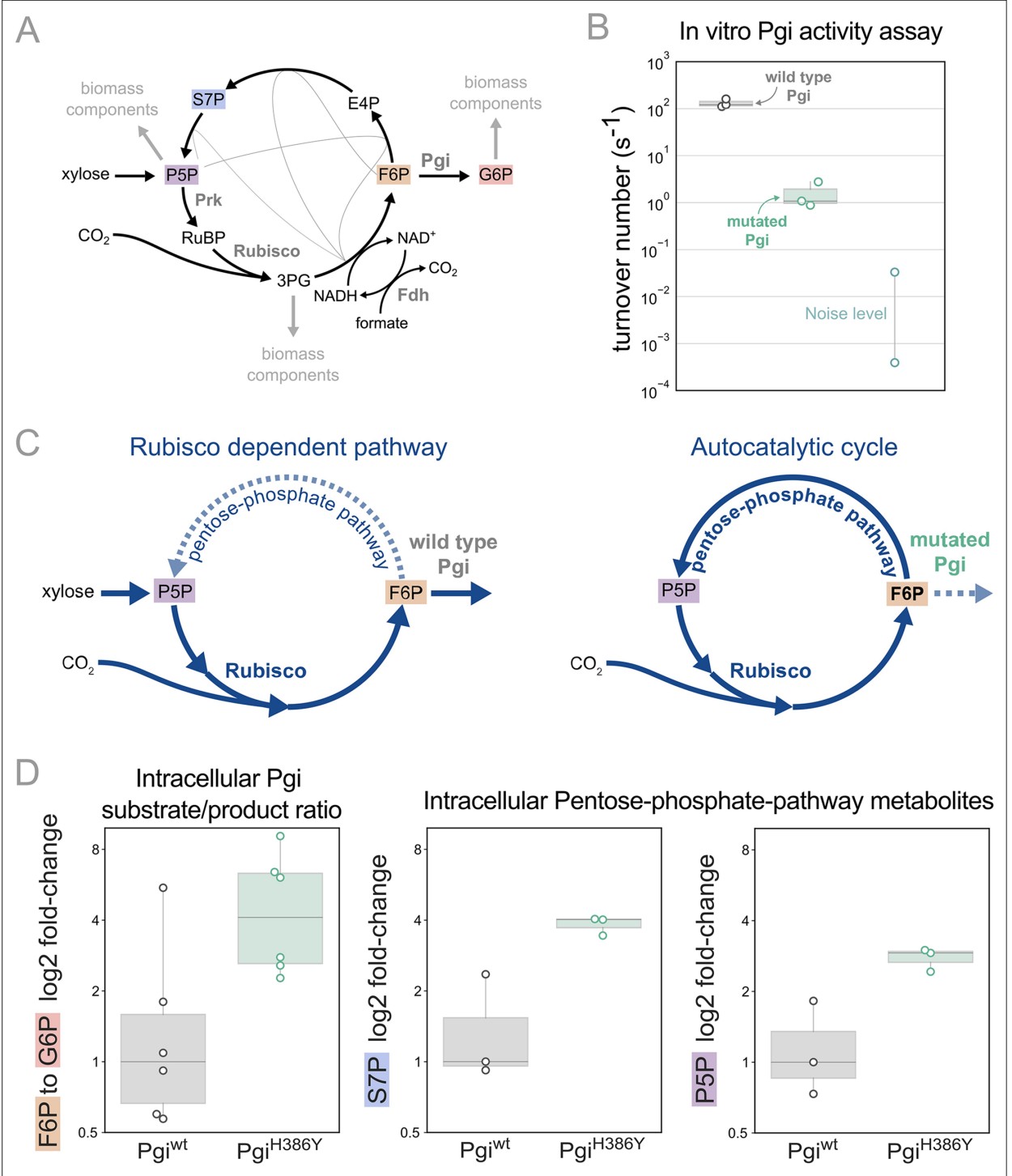

**Figure 4.** Pgi mutation is required to partition flux towards regeneration of autocatalytic cycle substrate. (**A**) Metabolic scheme of rPP cycle components when growing on xylose. (**B**) In vitro spectrophotometric coupled assay determined that the rate of the isomerization of fructose-6-phosphate (F6P) to glucose-6-phosphate (G6P) is ≈100-fold lower for purified Pgi[H386Y] (≈1.5 [s⁻¹]) compared to Pgi[wild-type] (≈130 [s⁻¹]), n=3. Noise level was determined by negative control samples containing a non-Pgi enzyme. (**C**) Left: The wild type Pgi, competes for its substrate F6P with the autocatalytic cycle, resulting in low F6P pool and low regeneration rate of ribulose-5-phosphate (Ru5P), which means that the RuBisCO-dependent pathway requires constant xylose supply. Right: The mutated Pgi (green) has reduced activity, which increases the ratio between the flux in the cycle and efflux which is needed for a stable regenerative flux towards Ru5P (via the pentose-phosphate-pathway), thus enabling an autotrophic cycle. (**D**) Left: Measured relative intracellular ratio of F6P and G6P of the ancestor background harbouring a Pgi[H386Y] mutation compared to the ancestor with a wild-type Pgi, both grown in RuBisCO-dependent conditions, n=6 cultures p-value <0.05. Right: Relative intracellular abundance of pentose-phosphate-pathway metabolites - sedoheptulose-

*Figure 4 continued on next page*

*Figure 4 continued*

7P (S7P) and total pool of pentose phosphates (ribulose-5P, ribose-5P, ribose-1P, and xylulose-5P - denoted P5P) in a Pgi$^{H386Y}$ strain versus the Pgi$^{wild-type}$ strain, both growing in a RuBisCO-dependent manner, n=3 cultures, p-value <0.05. p-values are based on Student's t-test with equal variance.

The online version of this article includes the following source data and figure supplement(s) for figure 4:

**Source data 1.** In vitro characterization of Pgi$^{H386Y}$.

**Figure supplement 1.** In vitro characterization of Pgi$^{H386Y}$.

**Figure supplement 2.** LC-MS/MS chromatogram of intracellular metabolites involved in the autocatalytic cycle in *E. coli* ancestor with Pgi$^{H386Y}$ mutation and ancestor with Pgi$^{wild-type}$.

(LC-MS/MS). For this comparison, we used the RuBisCO-dependent ancestor strain as the genetic background. We found that the ratio of F6P to G6P was about three times higher in the *pgi* mutant strain relative to the wild-type (*Figure 4D*). Furthermore, the *pgi* mutant had higher levels of metabolites within the rPP cycle (*Figure 4D*), confirming the stabilising function of the mutation.

These results lead us to conclude that the recurring mutations in the *pgi* gene are mediating the integration of the non-native carbon fixation genes into a stable autocatalytic cycle. This is achieved by reducing the ratio of the efflux from F6P to G6P relative to the flux in the cycle, thereby redirecting it towards the rPP cycle and the regeneration of RuBP.

## Proteomic analysis reveals up-regulation of rPP cycle and formate-associated genes alongside down-regulation of catabolic genes

The other two mutations in the autotrophy-enabling set, *crp* (H22N) and *rpoB* (A1245V), are global effectors in the cell. Crp is one of the most general transcription factors in the cell and has a role in many processes from metabolism to osmoregulation and biofilm formation (*Landis et al., 1999*; *Jackson et al., 2002*; *Ammar et al., 2018*). The position H22 in Crp is part of 'activation region 2' of the protein, an interaction region with RNA polymerase (*Figure 5—figure supplement 3*). This position was shown to have an effect on promoter specificity (*Niu et al., 1996*) and was previously reported to have a role in the integration of the rPP cycle in *E. coli* (*Herz et al., 2017*).

Similarly, mutations in *rpoB,* a subunit of RNA polymerase, have been shown to have a large effect on the cell transcriptome (*Conrad et al., 2010*; *Utrilla et al., 2016*). The observed A1245V mutation in *rpoB* is situated within the core of RNA polymerase, proximal to its interaction with the DNA (*Figure 5—figure supplement 3*). Furthermore, this genetic variation was previously documented in the adaptation of the *E. coli* B REL1206 strain to high temperatures (*Tenaillon et al., 2012*) and is closely positioned to another previously reported *rpoB* mutation (K1242Q) that appeared in adaptation to loss of Pgi activity (*Charusanti et al., 2010*).

Depending on the transcribed gene target, the effect of the two mutations might be additive, antagonistic, or synergistic. Since each one of these mutations individually (in combination with the *pgi* mutation) is not sufficient to achieve autotrophic growth, it is reasonable to assume that only the target genes whose levels of expression change significantly in the double-mutant are the ones relevant for the autotrophic phenotype.

Given the accumulated knowledge described above, these mutations are likely to have a broad effect on the cell. We chose to further elucidate their effect using a proteomics approach. As such, we conducted two experiments in an attempt to find changes in expression that are relevant for the adaptation to the new trophic mode. The first experiment compared the proteome of an evolved clone to its ancestor. The second experiment similarly compared an engineered BW25113 (BW) strain, containing the two regulatory mutations from the autotrophy-enabling set (i.e. *crp* H22N and *rpoB* A1245V) together with the deletions used to insert them (see Materials and methods and *Supplementary file 1*), to a 'wild type' BW strain (a corresponding knockout strain without the mutations, see Materials and methods).

Since our aim was to observe only direct effects induced by the genetic changes, rather than by the environment, we measured the proteomes of the strains under identical conditions and dilution rates. Ideally, the comparison should be done in conditions that require autotrophic growth (since some effects might depend on the lack of an organic carbon source), however, all strains except the evolved clone cannot grow in such conditions. Therefore, we decided on growing the cells in the conditions

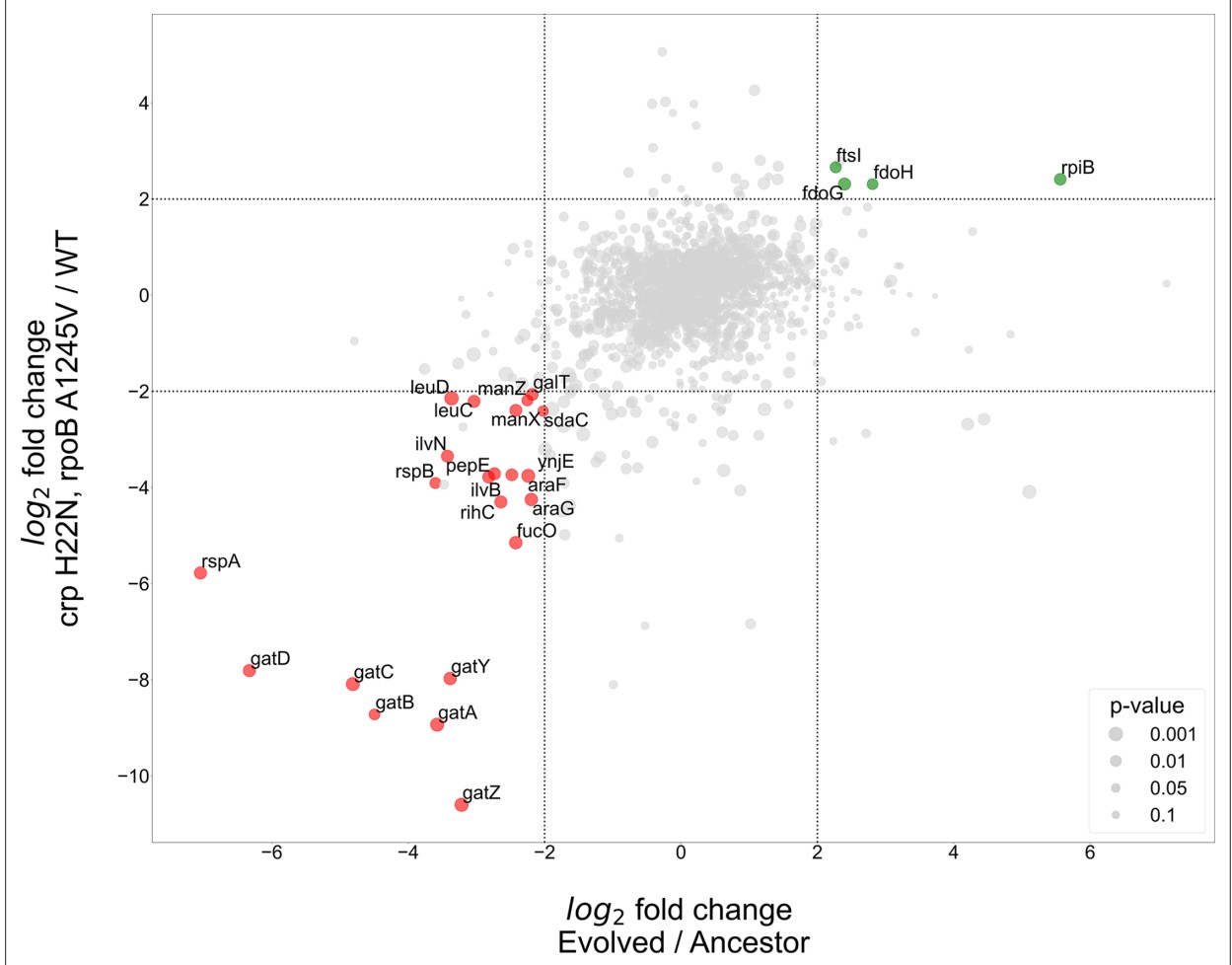

**Figure 5.** Proteomics analyses of fold change in evolved/ancestor expression levels versus double mutant/WT. Comparison of the log2 fold change in the evolved autotrophic strain divided by the ancestor expression versus BW rpoB A1245V and crp H22N, divided by the WT. The size of dots represents significance, only genes with >fourfold-change who showed significance (<0.05 p-value) were annotated.

The online version of this article includes the following figure supplement(s) for figure 5:

**Figure supplement 1.** proteomics analysis of evolved strain and BW double-mutant.

**Figure supplement 2.** proteomics analysis of BW mutants.

**Figure supplement 3.** 3D structure of wild type RNA polymerase and cAMP receptor protein.

present at the beginning of the lab evolution experiment, namely a xylose-limited chemostat. The dilution rate was set to ≈0.03 hr⁻¹ leading to a doubling time of ≈24 hr (Materials and methods).

In order to identify the genes with a large and significant fold change (fold change >4, p-value <0.05) we generated a volcano plot for each such comparison (*Figure 5—figure supplement 1*). We then proceeded by comparing the output of the first experiment (evolved divided by ancestor, *Figure 5—figure supplement 1A*) to that of the double mutant divided by the base BW strain (*Figure 5—figure supplement 1B*), since both mutations are essential for the phenotype and are known to affect expression. We focused on genes with common trends of expression level in both sets (*Figure 5*).

The proteomics analysis yielded few genes that had an upregulation of >fourfold in both the evolved versus ancestor and the double mutant versus BW base strain experiments. Two of these up-regulated genes, *fdoG* and *fdoH*, are subunits of the native formate dehydrogenase O enzyme complex (FDH-O) and are therefore related to the energy module. These genes were upregulated in both the evolved strain and in the strain with the *crp* H22N and *rpoB A1245V* double mutation (≈five-fold increase in both cases). The FDH-O enzyme complex takes part in the respiratory chain formed

by formate dehydrogenase and dimethyl sulfoxide (DMSO) reductase. Here, the transfer of electrons from formate to DMSO (via a quinone electron carrier) is coupled to the generation of a proton-motive force across the cytoplasmic membrane for energy formation.

Another observed upregulated protein was ribose-phosphate-isomerase B (*rpiB*), which was up-regulated in the evolved strain (>40-fold increase) and in the double mutant ≈fivefold increase (*Figure 5*). *rpiB* is an inducible isoenzyme of the constitutively expressed ribose-phosphate-isomerase A, a core enzyme in the rPP cycle. This might serve as evidence for a higher flux requirement through the pentose-phosphate-pathway, supported by expressing a second isoenzyme. Other explanations may include some form of regulation, perhaps allosteric, inhibiting RpiA activity, making the strain dependent on RpiB activity, or non-specific enzymatic activity of RpiB which is required by the cell. The fourth up-regulated gene - *ftsI*, a peptidoglycan DD-transpeptidase - does not have a clear explanation at this time.

At the other end of the expression spectrum, several pathways were down-regulated in both the evolved and in the BW double mutant strain (*Figure 5*). Down-regulation of the starvation response, metabolic and PTS component genes seems to be at the heart of the adaptation to autotrophic growth, with several operons experiencing >fourfold decrease in expression. Most notably the *gat*, *ara* and *man* operons responsible for galactitol, arabinose and mannose degradation, respectively. Again, this is evident in both the evolved and the double mutant strains (*Figure 5*). Other highly down-regulated metabolic pathways include amino acid biosynthesis, specifically valine and leucine, moreover we observed a down regulation in *rspA and rspB*, which encode for starvation sensing proteins. We suggest these results mean that many pathways usually up-regulated in *E. coli* during starvation may be redundant or even counter-productive in an autotrophic context, and decreasing their expression could result in increased fitness.

We additionally tested whether separating between the two regulatory mutations (*crp* and *rpoB*) might provide a clearer picture, for example if different up- or down-regulated genes are targeted by only one of these mutations and not the other. Therefore, we used single-mutant BW strains and compared them to the same BW wild-type as before. Essentially, the results did not change our conclusions and although in most cases the effect of the two mutations was cumulative, we did find at least one case where they were antagonistic (*Figure 5—figure supplement 2*). For example, *fdoG* and *fdoH* showed a ≈fourfold *decrease* compared to wild type when the *crp* mutation was introduced, but (as stated earlier) had a ≈fivefold abundance *increase* in the double mutant, suggesting an epistatic effect between the *rpoB* and *crp* mutations (*Figure 5—figure supplement 1B* and *Figure 5—figure supplement 2B*). All the comparisons based on the single and double mutant strains can be found in the supplementary.

## Mutations in regulatory genes lead to increased availability of the carbon fixation cycle electron donor NADH

Along with the effect on expression, we suspected the two transcription-associated mutations (*crp* and *rpoB)* to also have a metabolic and physiological effect. Since Crp and RpoB are both essential to the autotrophic phenotype and are known to indirectly interact in the cell (*Savery et al., 1996*; *Niu et al., 1996*; *Frendorf et al., 2019*), we address them as one unit. The proteomics analysis, where the strain with both mutations seem to have a mutual effect that is in some cases different than in the individual mutants, supported this decision. Mutations in *crp* and *rpoB* are known to affect growth at different conditions such as types of media or carbon sources (*Utrilla et al., 2016*; *Conrad et al., 2010*; *Cheng et al., 2014*; *Frendorf et al., 2019*, *Aidelberg et al., 2014*). Therefore, we tested whether this was also the case for the conditions under which the cells evolved in the chemostat.

We performed a growth experiment of the double mutant (*crp* H22N, *rpoB* A1245V) expressing *fdh* in a minimal medium containing xylose and formate. Under such conditions, the strain harbouring both mutations and *fdh* exhibited >25% increase in the final OD compared to the wild type expressing *fdh* (*Figure 6A&B*). At the same time they also showed a longer (>10 hr) lag time. Notably, when grown solely on xylose, in the absence of formate, the increase in yield was not observed, and the extended lag was less pronounced (*Figure 6A*).

Since NADH is the electron donor for carbon fixation, and a product of formate consumption via Fdh, we used metabolomics to measure the ratio of NADH/NAD$^+$ in these strains in similar conditions (Materials and methods). All of the measured ratios were normalised to the wild-type (with *fdh*)

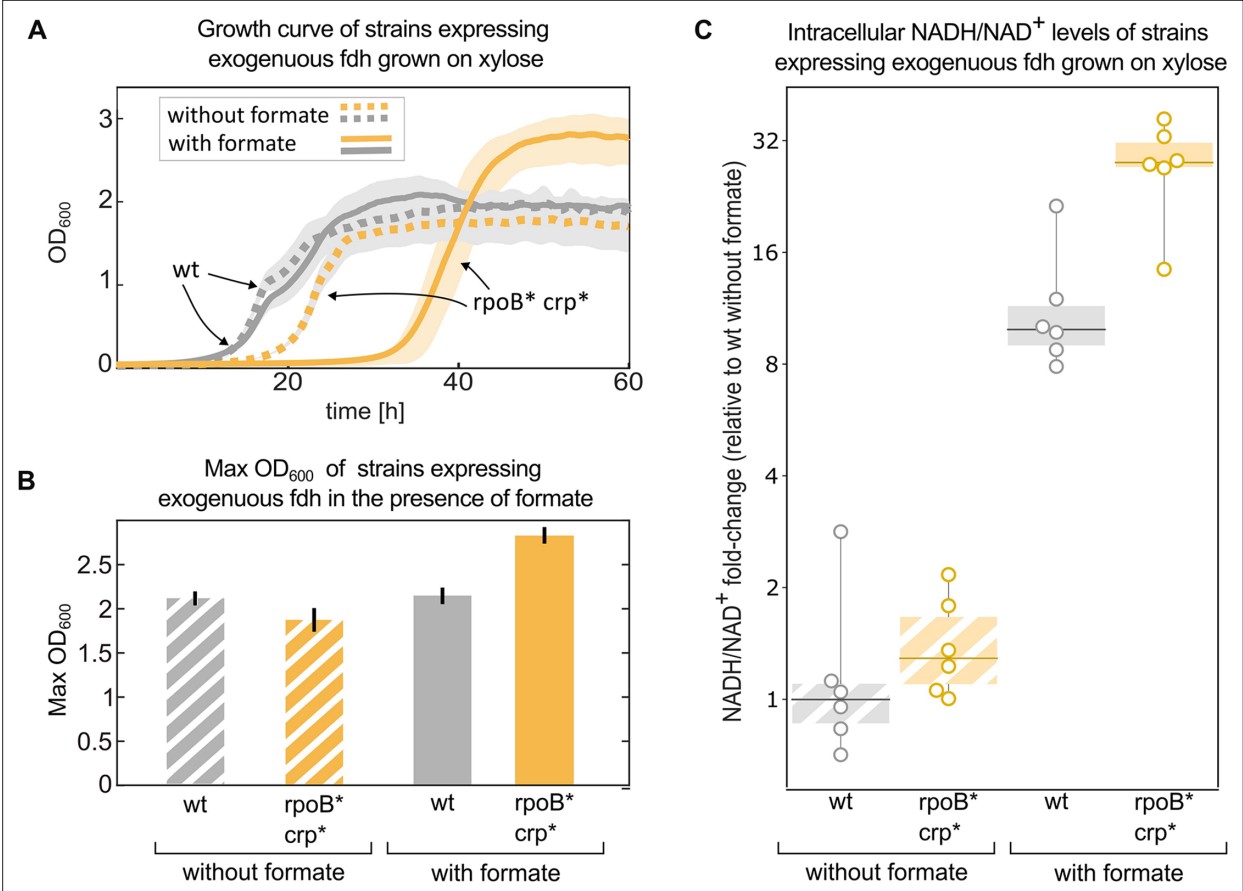

**Figure 6.** Mutations in *rpoB* and *crp* increase yield and intracellular NADH/NAD $^+$ levels in fdh-expressing *E. coli* in the presence of formate. Growth experiment of BW25113 wild-type (grey) compared to a *crp* H22N *rpoB* A1245V mutant (orange). Both strains express *fdh*. (**A**) Strains were grown in 1 g/L xylose and 40 mM formate (solid line) or in the absence of formate (dashed line). The experiment was executed in a 96-well plate in 10% $CO_2$ atmosphere, n=6 repeats. Lines represent the mean, light-grey background represents the standard deviation of the mean. (**B**) Maximal $OD_{600}$ of the wild-type (grey) and mutant (orange) strains grown in 1 g/L xylose and 40 mM formate or in the absence of formate. Bar heights represent averages (n=6) of the median of the top 10 OD measurements of each replicate. Error bars represent standard deviation of the mean. (**C**) Intracellular NADH/ $NAD^+$ ratio of the wild-type (grey) was compared to the mutant (orange). The strains were grown in 2 g/L xylose and 30 mM formate, or in the absence of formate. The y-values are NADH/$NAD^+$ ratios as fold-changes relative to wild-type without formate. Boxes represent 25–75 percentile ranges and dark lines represent median values. All data points are depicted without removing outliers. n=6 cultures, for the comparison in the presence of formate p-value <0.05. p-values are based on Student's t-test with equal variance.

The online version of this article includes the following figure supplement(s) for figure 6:

**Figure supplement 1.** LC-MS/MS chromatograms of NADH and NAD in fdh-expressing *E. coli* in the presence and absence of formate.

growing in the absence of formate. The addition of 30 mM formate, which can serve as an electron donor via oxidation to $CO_2$ by Fdh, increased NADH/$NAD^+$ ratio by>20-fold in the double mutant, significantly higher than the ≈10-fold increase in the wild-type strain (p-value <0.05). No significant difference was observed between the strains in the absence of formate (*Figure 6C*).

NADH could affect growth in several ways, as discussed below. The specific reason for the increase in final OD or lag time remains inconclusive at this point but we hypothesise that it is related to the increased expression of Fdo observed in the proteomic analysis. We also note that as the proteomics results suggest - the mutations in global transcriptional machinery like Crp and RpoB could have many other effects on the autotrophic phenotype (*Haverkorn van Rijsewijk et al., 2011*) which await future exploration.

## Discussion

In this study, we demonstrated how iterative evolution and engineering were able to narrow down the genomic mutations required for autotrophy in *E. coli* from tens of mutations to only three mutated genes. The three mutated genes - *pgi*, *crp* and *rpoB* - serve as a compact set, which when introduced into a wild-type background and supplemented with the carbon-fixing and energy modules, achieve an autotrophic phenotype. By metabolic, proteomic and physiological characterisation we were able to distil some of the effects of these mutations and understand their role in facilitating the autotrophic phenotype. We suggest that the three main effects achieved by the autotrophy-enabling mutations are: (i) integrating the non-native carbon fixation cycle by enhancing flux within the cycle over outgoing flux; (ii) facilitating the integration of the energy module via decoupling of formate harvesting into energy and redox, thereby increasing the NADH/NAD$^+$ ratio, which could have both thermodynamic and kinetic benefits (iii) and freeing protein resources by decreasing unneeded expression.

The mutation in *pgi* affects the activity of its gene product phosphoglucoisomerase - a metabolic enzyme that catalyses a reaction that branches out of the non-native rPP cycle. The enzyme converts F6P (a cycle intermediate) to G6P (a precursor required for membrane biosynthesis). In a previous study, this mutation was found to be essential for the integration of the rPP autocatalytic cycle and was part of a minimal set for hemi-autotrophic growth (*Gleizer et al., 2019*; *Gassler et al., 2020*; *Antonovsky et al., 2016*; *Barenholz et al., 2017*; *Flamholz et al., 2020*; *Herz et al., 2017*). As our results suggest, the mutated enzyme shows a significant reduction in activity, which decreases the efflux from the rPP cycle to biosynthesis. We hypothesised that its role is to generate a new, higher, steady-state concentration for the rPP cycle metabolites, thus stabilising the cycle. This was supported by our observation that introducing a *pgi* mutation into a metabolically rewired RuBiCO-dependent strain resulted in increased concentration of rPP cycle intermediates. This exemplifies the key role branch-points play in autocatalytic cycles (*Gleizer et al., 2019*; *Gassler et al., 2020*; *Antonovsky et al., 2016*; *Barenholz et al., 2017*; *Flamholz et al., 2020*; *Herz et al., 2017*), making them a leading target for rational attempts to integrate non-native autocatalytic cycles into metabolism.

The remaining two mutations act as global regulators, which presents a challenge in distinguishing their specific adaptive impacts from their broader effects on cellular expression patterns. Using proteomic analyses, we have managed to identify certain genes that had a major change in expression when one of the regulating mutations was introduced, and when both were introduced together.

One main observation was the down-regulation of several metabolic pathways including catabolism of alternative carbon sources and amino acid biosynthesis. We suggest this is an adaptive role in improving cell resource allocation, and a means to modulate resource allocation, possibly freeing up proteome resources from alternative carbon metabolism proteins. This includes proteins such as PTS system components or ATP dependent transporters, that tend to incur high energetic costs.

Other reasons these catabolic and transport pathways may result in reduced fitness could include competition over metabolites with the rPP cycle, or by causing other metabolic blunders such as dead-end products with potential toxicity effects under the new flux regime. Another observed effect the *crp* mutation shows is the up-regulation of *rpiB*, a typically unexpressed isoenzyme to one of the reactions in the rPP cycle. This expression increase could be beneficial or even essential in a scenario where *rpiA*, the main enzyme catalysing this reaction, cannot support the flux required by the rPP cycle, for example due to kinetic capacity limitations or allosteric inhibition. Further work is needed to investigate if up-regulation of *rpiB* is indeed essential for the autotrophic phenotype.

We also suggest these mutations play a role in improving the utilisation of formate, which is supported by the up-regulation of *fdoG & fdoH*. We found that a strain introduced with both the *crp* and *rpoB* mutations demonstrated a pronounced up-regulation of over 4-fold and also a concrete fitness advantage in growth conditions which include formate. Metabolomics analysis showed this strain also has a significant increase in the NADH/NAD$^+$ ratio. NADH is the essential cofactor for the electron transfer reaction of the non-native rPP cycle (catalysed by the D-glyceraldehyde-3P dehydrogenase GAPDH) and a product of the non-native formate-based energy module. The importance of this up-regulation is also supported by natural chemolithoautotrophic metabolism, which uses two distinct ways to harvest energy and reducing power from the electron-donor. For example, in the model chemolithoautotroph *Cupriavidus necator*, the oxidation of formate is catalysed by two distinct types of FDH: a soluble, NAD$^+$-linked enzyme (S-FDH; EC 1.2.1.2) and a membrane-bound enzyme coupled directly to the respiratory chain (*Friedrich et al., 1979*; *Oh and Bowien, 1998*).

We suspect utilising formate dehydrogenase O (FDH-O) in the periplasm is enabling the cell to also utilise formate directly on the membrane to create a proton motive force for energy production without the need to exhaust the intracellular NADH pool, thus allowing for its accumulation. Increased levels of NADH creates favourable thermodynamics for the rPP cycle flux and could also offer improved kinetics. This result is in line with previous models, predicting that high cofactor levels are required for a stable metabolic state of the rPP cycle (*Janasch et al., 2019*). We suggest this observation offers an avenue for future rational design attempts.

We continue by hypothesising that such regulatory mutations are to be expected in adaptive laboratory evolution experiments, especially during selection for a major shift (novelty) in the phenotype. A major shift will often necessitate a modulation of the activity of many cellular components. Lab evolution is limited by population size and number of generations, which means it can only explore a small part of the genotypic space. Therefore, sets of three or more mutations, where each mutation on its own does not confer an advantage, have a negligible probability to occur. On the other hand, a single mutation in a global regulator could concurrently perturb the activity of hundreds of components and might be the only viable solution to capture several required changes simultaneously.

Another reported observation is how metabolic rewiring can serve as a 'metabolic scaffold' to direct lab evolution. Since scaffold mutations are put in place for the purposes of selection, they can be removed once the goal is achieved. A common limitation of lab evolution is its tendency to follow evolutionary trajectories which increase fitness by deactivating or losing the heterologously expressed genes, thereby arriving at 'dead-end' local optimum. A 'metabolic scaffold' as utilised here can block these dead-end trajectories by coupling a target activity (here RuBisCO carboxylation) to growth, thereby increasing the likelihood of trajectories that lead to the desired phenotype. Once the phenotype is achieved, removal of the metabolic scaffold should be possible since growth is by definition dependent on the non-native genes. Using metabolic growth-coupling as a temporary 'metabolic scaffold' that can be removed, could serve as a promising strategy for achieving minimally perturbed genotypes in future metabolic engineering attempts.

The approach presented here, of harnessing iterative lab evolution to find a compact set of essential mutations, strikingly shows the extreme malleability of metabolism and evolution's capacity to switch trophic modes in laboratory time-scales with very few mutations. It can also serve as a promising pipeline for other genetic engineering attempts. Having a compact and well-defined genotype in a model organism allowed us to reveal metabolic adaptation steps needed for autotrophic growth, and could be used for future exploration of the design principles of the rPP cycle operation and its components (*Aigner et al., 2017*). The harnessing and rationalisation of the lab evolution process and metabolic adaptation steps elucidated here could facilitate the introduction of the rPP cycle and other carbon fixation cycles into other organisms of interest. Next steps could include integrating metabolic bio-production modules and alternative energy sources. Now that synthetic autotrophy in *E. coli* can be achieved with only a few defined steps, it expands the playground for more labs to join the quest for sustainable bio-production from $CO_2$.

# Materials and methods
## Strains
We generated an engineered ancestor strain for chemostat evolution based on the *E. coli* BW25113 strain (*Grenier et al., 2014*). We used P1 transduction (*Thomason et al., 2007*) to transfer knockout alleles from the KEIO strain collection (*Thomason et al., 2007*; *Baba et al., 2006*) to our engineered strain, and to knock out the genes phosphofructokinase (*pfkA* and *pfkB*), and glucose-6-phosphate dehydrogenase (*zwf*). Following the transduction of each knockout allele the Kanamycin resistance (KmR) selection marker was removed by using the FLP recombinase encoded by the pCP20 temperature-sensitive plasmid (*Cherepanov and Wackernagel, 1995*). Loss of the selection marker and the temperature sensitive plasmid were validated by replica-plating the screened colonies and PCR analysis of the relevant loci. The cells were then transformed with the pCBB plasmid(accession number KX077536) and a pFDH plasmid with a constitutive promoter controlling the expression of the fdh gene that resulted from *Gleizer et al., 2019*. The transfer of mutations from evolved clones for the generation of ancestor 2.0, and other mutated strains was also accomplished by P1 transduction as reported in *Herz et al., 2017*. P1 lysate was prepared from Keio collection strains that had a

Kanamycin (Km) resistance marker located near the desired allele (as specified in *Supplementary file 1*). The strains carrying the desired mutated allele were transduced with P1 phages containing the resistance marker and then plated on Km plates. Donor strains were selected based on the presence of the desired mutated allele and the Km resistance marker in their colonies. A P1 lysate from each donor strain was used to transfer the desired mutations to a recipient strain.

Because the transferred DNA fragment to the recipient was relatively large (approximately 100 kb), including the selection marker, it often carried adjacent genes. Genotyping was performed using Sanger sequencing. For further genomic modification, pCP20, a temperature-sensitive plasmid encoding the FLP recombinase from the Clone and Strain Repository unit at WIS, was used following standard procedures. This allowed for the removal of Km resistance markers, enabling iterative rounds of genome editing.

## Plasmids

To create the pFDH plasmid, an *E. coli* codon optimised DNA sequence of formate dehydrogenase from the metholotrophic bacterium *Pseudomonas sp. 101* (*Popov and Lamzin, 1994*) was cloned with an N-terminal His-tag into a pZ plasmid (Expressys, Germany) under a constitutive promoter and with a strong ribosome binding site (rbs B of *Zelcbuch et al., 2013*). The plasmid has a pMB1 origin of replication and therefore is present in high copy number. We replaced the KmR selection marker on the plasmid with a StrepR marker. An 8 bp deletion appeared in the promoter region of the first evolved clone. This plasmid was isolated and was the plasmid used for all consequential evolution experiments and autotrophic strains (*Gleizer et al., 2019*). Details regarding the pCBB plasmid are reported in *Antonovsky et al., 2016*.

## Growth tests

The growth test experiments were conducted in 96 well-plates. The final volume of each well was 200 μL (50 μL of mineral oil and 150 μL culture). The media consisted of M9 media supplemented with varying concentrations of the relevant carbon source, and trace elements (without addition of vitamin B1). Bacterial cells were seeded from a culture tube. Growth temperature was set to 37 °C, and either aerated with ambient air or air with elevated $CO_2$ (10%) either with ambient or reduced oxygen (5%). OD600 measurements were taken every 5–30 min using a Tecan Spark plate reader.

## Chemostat evolution experiments

The evolutionary experiment was conducted in a Bioflo 110 chemostat (New Brunswick Scientific, USA) at a working volume of 0.7 L and a dilution rate of 0.02 h$^{-1}$ (equivalent to a doubling time of ≈33 hr) at 37 °C. The chemostat was fed with media containing 4 g/L sodium formate and 0.5 g/L D-xylose as sole carbon sources. This amount of xylose in the feed makes xylose the limiting nutrient for cell growth in the chemostat. Gradually the concentration of xylose in the feed was reduced until a phenotype was observed. Chloramphenicol (30 mg/L) and streptomycin (100 mg/L) were added to the feed media. Aeration of the chemostat was done through a DASGIP MX4/4 stand-alone gas-mixing module (Eppendorf, Germany) with a composition of 10% $CO_2$ and 5% oxygen and 85% air at a flow rate of 40 sL/hr. To monitor the chemostat, a weekly sampling protocol was performed. Samples were taken for media analysis and phenotyping (inoculation of the bacteria on minimal media containing formate and lacking D-xylose). The optical density of each extracted sample was measured using a spectrophotometer (Ultrospec 10 Cell density meter, Amersham Biosciences) and a standard 10 mm polystyrene cuvette (Sarstedt, Germany).

## $^{13}$C isotopic labeling experiment

A culture of cells that were growing on naturally labeled sodium formate in an elevated $CO_2$ (10%, naturally labeled) incubator (New Brunswick S41i $CO_2$ incubator shaker, Eppendorf, Germany) were diluted 10-fold into fresh M9 media with 30 mM $^{13}$C-formate sodium salt (Sigma Aldrich) to a total volume of 10 ml of culture. In the 'open' labeling setup, growth was carried out in 125 ml glass shake flasks (n=3), which allow free exchange of gases between the headspace of the growth vessel and the gas mixture of the incubator. The flasks were placed inside an elevated $CO_2$ (10%) shaker-incubator (New Brunswick) with 37 °C. After ≈4 doublings, the cultured cells were combined and harvested for subsequent analysis of protein bound amino acids and intracellular metabolites. In the 'closed'

labeling setup, growth was carried out in 250 mL glass shake flasks with a transparent extension which allows measurement of the optical density of the culture without opening it. After ≈3 doublings, the cells were diluted eightfold into flasks covered with air-tight rubber septa (SubaSeal, Sigma-Aldrich). Then, the headspace of the flask was flushed with a gas mixture containing 10% $^{13}CO_2$ (Cambridge Isotope Laboratories, USA)+90% air or 10% $^{12}CO_2$ +90% air generated by a DASGIP MX4/4 stand-alone gas-mixing module (Eppendorf, Germany). The flasks were placed in a 37 °C shaker incubator. After ≈4 doublings, the cells were harvested for subsequent analysis of protein bound amino acids. Glass flasks used in the labeling experiments were pretreated by heating in a 460 °C furnace for 5 hr to evaporate any excess carbon sources that could remain in the vessels from previous utilizations. Each labeling experiment was conducted in triplicates (n=3), with $^{12}CO_2$ the triplicates were pooled and measured as three technical replicates. The predicted range of labeling under $^{13}CO_2$ was calculated using the following equation: $f_n = 1 - 0.5^n$ , where f is the $^{13}C$ fraction and n is the number of doublings, as the remaining unlabeled biomass is decreased twofold at each doubling. The $^{13}C$ fraction was calculated for 3 and 5 doublings to create the predicted range.

## Growth comparison between engineered autotrophic strains

The growth was conducted in a DASBox mini fermentation system (Eppendorf, Germany). The starting volume of each bioreactor was 150 ml M9 media supplemented with 60 mM Formate. The minimal media also included trace elements, vitamin B1 was omitted. The atmosphere was elevated 10% $CO_2$ 5% $O_2$ and 85% $N_2$. The culture was grown to saturation and then diluted by a ratio of 1:30, to obtain technical replicates.

## Sample preparation for liquid chromatography coupled to mass spectrometry and mass analysis of biomass components

After harvesting the biomass culture samples were prepared and analysed as described in **Antonovsky et al., 2016**. Briefly, for protein bound amino acids ≈3 mL of culture at OD600 turbidity of ≈0.1–0.15 were pelleted by centrifugation for 5 min at 8000 × g. The pellet was suspended in 1 mL of 6 N HCl and incubated for 16 hr at 110 °C. The acid was subsequently evaporated with a nitrogen stream, resulting in a dry hydrolysate. Dry hydrolysates were resuspended in 0.6 mL of miliQ water, centrifuged for 5 min at 14,000 × g and the supernatant was used for injection into the LCMS. Hydrolysed amino acids were separated using ultra performance liquid chromatography (UPLC, Acquity - Waters, USA) on a C-8 column (Zorbax Eclipse XBD - Agilent, USA) at a flow rate of 0.6 mL/min and eluted off the column using a hydrophobicity gradient. Buffers used were: (A) $H_2O$+0.1% formic acid and (B) acetonitrile +0.1% formic acid with the following gradient: 100% of A (0–3 min), 100% A to 100% B (3–9 min), 100% B (9–13 min), 100% B to 100% A (13–14 min), 100% A (14–20 min). The UPLC was coupled online to a triple quadrupole mass spectrometer (TQS - Waters, USA). Data was acquired using MassLynx v4.1 (Waters, USA). We selected amino acids which have peaks at distinct retention time and m/z values for all isotopologues and also showed correct $^{13}C$ labeling fractions in control samples that contained protein hydrolyzates of WT cells grown with known ratios of $^{13}C6$-glucose to $^{12}C$-glucose.

The $^{13}C$ fraction of each metabolite was determined as the weighted average of the fractions of all the isotopologues of the metabolite, as depicted in the equation below:

$$\%^{13}C = \frac{\sum_{i=0}^{n} f_i \cdot i}{n}$$

where n is the number of carbons in the compound (e.g., for the amino acid serine n=3) and $f_i$ is the relative fraction of the i-th isotopologue.

## Whole-genome sequencing

DNA extraction (DNeasy blood & tissue kit, QIAGEN) and library preparation procedures were carried out as previously described in **Herz et al., 2017**; **Antonovsky et al., 2016**. The prepared libraries were sequenced by a Miseq machine (Illumina). Analysis of the sequencing data was performed as previously described in **Herz et al., 2017**; **Antonovsky et al., 2016**; **Gleizer et al., 2019** with the breseq software (**Barrick et al., 2014**).

## Cultivation conditions for metabolome sampling

For the G6P and F6P, P5P, and S7P measurements: Single colonies were transferred into liquid 3 mL M9 media containing 0.2 g/L xylose, 2 g/L formate, Chloramphenicol (30 mg/L) and strepto-mycin (100 mg/L) from fresh plates with the same medium. The M9 pre-cultures were adjusted to a starting OD600 of 0.05 into 10 mL culture in 250 mL glass shake flasks. The flasks were placed inside an elevated $CO_2$ (10%) shaker-incubator (New Brunswick S41i $CO_2$ incubator shaker, Eppendorf, Germany) at 37 °C.

For NADH/NAD+ measurements: Single colonies were transferred into 10 mL M9 xylose strepto-mycin from fresh M9 xylose streptomycin plates, and cultivated for 24 hr while shaking at 37 °C. M9 pre-cultures were adjusted to a starting OD600 of 0.1 into 12-well plates, with 2 mL of medium in each well. Strains were cultivated in triplicates with or without formate (30 mM), added at the beginning of the culture. Optical density at 600 nm was monitored every 10 min using a plate reader (Tecan, Spark). Plates were then rapidly transferred to a thermostatically controlled hood at 37 °C and kept shaking during the sampling procedure.

## Metabolomics measurements

Cultivations were performed as described above. Culture aliquots were vacuum-filtered on a 0.45 µm pore size filter (HVLP02500, Merck Millipore). Filters were immediately transferred into a 40:40:20 (v-%) acetonitrile/methanol/water extraction solution at –20 °C. Filters were incubated in the extraction solution for at least 30 min. Subsequently, metabolite extracts were centrifuged for 15 min at 13,000 rpm at –9 °C and the supernatant was stored at –80 °C until analysis. For measure-ments of NADH, NAD+, sedoheptulose-7P and the pool of pentose phosphates, extracts were mixed with a $^{13}C$-labeled internal standard in a 1:1 ratio. LC-MS/MS analysis was performed with an Agilent 6495 triple quadrupole mass spectrometer (Agilent Technologies) as described previously (**Donati et al., 2021**). An Agilent 1290 Infinity II UHPLC system (Agilent Technologies) was used for liquid chromatography. Temperature of the column oven was 30 °C, and the injection volume was 3 µL. LC solvents in channel A were either water with 10 mM ammonium formate and 0.1% formic acid (v/v) (for acidic conditions, NADH/NAD+), or water with 10 mM ammonium carbonate and 0.2% ammonium hydroxide (for basic conditions, sedoheptulose-7P and pool of pentoses). LC solvents in channel B were either acetonitrile with 0.1% formic acid (v/v) (for acidic conditions) or acetonitrile without addi-tive (for basic conditions). LC columns were an Acquity BEH Amide (30x2.1 mm, 1.7 µm) for acidic conditions, and an iHILIC-Fusion(P) (50x2.1 mm, 5 µm) for basic conditions. The gradient for basic and acidic conditions was: 0 min 90% B; 1.3 min 40% B; 1.5 min 40% B; 1.7 min 90% B; 2 min 90% B. The ratio of $^{12}C$ and $^{13}C$ peak heights was used to quantify metabolites. $^{12}C/^{13}C$ ratios were normalized to OD at the time point of sampling. For the measurements of intracellular glucose-6-P and fructose-6-P, metabolic extracts were 10 x concentrated using a vacuum evaporation (Eppendorf Concentrator plus) and resuspended in 40:40:20 (v-%) acetonitrile/methanol/water extraction solution. Temperature of the column oven was 30 °C, and the injection volume was 5 µL. LC solvent in channel A was water with 10 mM ammonium formate and 0.1% formic acid (v/v), and in channel B was acetonitrile with 0.1% formic acid (v/v). The LC column was a Shodex HILICpak VG-50 2D (2.0x150 mm). The gradient was: 0 min 90% B;3min 30% B; 58 min 30% B; 61min 90% B;63 min 90% B. The $^{12}C$ peak heights were used to quantify metabolites. Retention time of hexose-phosphates were determined with authentic standards of glucose-6-P.

## Chemostat proteomics experiment

The growth was conducted in a DASBox mini fermentation system (Eppendorf, Germany). The starting volume of each bioreactor was 150 ml M9 media supplemented with either 30 mM and 10 mM D-xy-lose for the evolved and ancestor samples or only 10 mM D-xylose for BW including compact set mutations and/or the deletions used for their insertions (**Supplementary file 1**). The minimal media also included trace elements, vitamin B1 was omitted. In experiments including the evolved strain and the ancestor strain, the atmosphere had elevated $CO_2$ (10%), and the chemostat dilution rate was set to 0.03 h$^{-1}$ corresponding to a doubling time of 24 hr.

## Proteomics analysis

The cell pellets were lysed and subjected to in-solution tryptic digestion using the S-Trap method (by Protifi). The resulting peptides were analyzed using nanoflow liquid chromatography (nanoAcquity)

coupled to high resolution, high mass accuracy mass spectrometry (Thermo Q Excative HFX). Each sample was analyzed on the instrument separately in discovery mode in a random order. Raw data was processed with MaxQuant v1.6.6.0. The data was searched with the Andromeda search engine against the *E. coli* BW25113 (K12) protein database as downloaded from Uniprot, appended with the mutant protein sequences and common lab protein contaminants. Search parameters included the following modifications: Fixed modification- cysteine carbamidomethylation. Variable modifications- methionine oxidation, and asparagine and glutamine deamidation. The quantitative comparisons were calculated using Perseus v1.6.0.7. Decoy hits were filtered out and only proteins that were detected in at least 2 replicates of at least one experimental group were kept.

## Data, materials, and software Availability

All data and code are stored in a publicly available GitLab repository (https://gitlab.com/milo-lab-public/compact-autocoli copy archived at *Milshtein et al., 2024*).

## Acknowledgements

We thank Alon Barshap, Alon Savidor, Aliza Fedorenko, Avi Flamholz, Avihu Yona, Daria Fedorova, Emanuel Avrahami, Ifat, Goldstein, Ilana Rogachev, Ido Cohen, Lior Greenspoon, Lior Shachar, Margarita Gortikov, Merav Hagag, Niv Antonovsky, Ofir Shechter, Ron Sender, Samuel Lovat, Tasneem Bareia, Tali Wiesel, Yafit Sugas, Yinon Bar-on, Yotam David, Yuval Kushmaro and Yuval Rosenberg for their support of this project. We acknowledge the De Botton Protein Profiling institute of the Nancy and Stephen Grand Israel National Center for Personalized Medicine, Weizmann Institute of Science for their services and contribution to this project. This research was generously supported by the Mary and Tom Beck Canadian Center for Alternative Energy Research, the Schwartz-Reisman Collaborative Science Program, the Ullmann Family Foundation and the Yotam Project. Prof. Ron Milo is the Head of the Mary and Tom Beck Canadian Center for Alternative Energy Research and the incumbent of the Charles and Louise Gartner Professorial Chair. RBN is a Weizmann SAERI fellow. EM is a fellow of the Ariane de Rothschild Women Doctoral Program. HL and VP acknowledge funding from the Cluster of Excellence EXC 2124 from the Deutsche Forschungsgemeinschaft. Funding: Mary and Tom Beck Canadian Center for Alternative Energy Research (RM), Schwartz-Reisman Collaborative Science Program (RM), Ullmann Family Foundation and the Yotam Project (RM), Sustainability and Energy Weizmann Doctoral Fellowship (RBN), Ariane de Rothschild Women Doctoral Program (EM), Cluster of Excellence EXC 2124 from the Deutsche Forschungsgemeinschaft (HL, VP)

## Additional information

### Competing interests

Roee Ben Nissan, Shmuel Gleizer, Elad Noor, Ron Milo: We declare the following provisional patent related to the manuscript, "An Engineered Autotrophic E. coli Strain for CO2 Conversion to Organic Materials." European Patent Office: EP4051695A1. The other authors declare that no competing interests exist.

### Funding

| Funder | Grant reference number | Author |
| --- | --- | --- |
| Deutsche Forschungsgemeinschaft | EXC 2124 | Vanessa Pahl Hannes Link |
| Mary and Tom Beck Canadian Center for Alternative Energy Research | | Ron Milo |
| Schwartz-Reisman Collaborative Science Program | | Ron Milo |

| Funder | Grant reference number | Author |
|---|---|---|
| Ullmann Family Foundation and the Yotam Project | | Ron Milo |
| Sustainability and Energy Weizmann Doctoral Fellowship | | Roee Ben Nissan |
| Ariane de Rothschild Women Doctoral Program | | Eliya Milshtein |
| Cluster of Excellence EXC 2124 from the Deutsche Forschungsgemeinschaft | | Hannes Link Vanessa Pahl |

The funders had no role in study design, data collection and interpretation, or the decision to submit the work for publication.

## Author contributions

Roee Ben Nissan, Eliya Milshtein, Conceptualization, Data curation, Software, Formal analysis, Funding acquisition, Validation, Investigation, Visualization, Methodology, Writing – original draft, Project administration, Writing – review and editing; Vanessa Pahl, Data curation, Software, Formal analysis, Funding acquisition, Validation, Investigation, Visualization, Methodology, Writing – original draft, Writing – review and editing; Benoit de Pins, Data curation, Software, Formal analysis, Validation, Investigation, Visualization, Methodology, Writing – original draft, Writing – review and editing; Ghil Jona, Shmuel Gleizer, Investigation, Methodology; Dikla Levi, Hadas Yung, Noga Nir, Dolev Ezra, Investigation; Hannes Link, Resources, Formal analysis, Supervision, Funding acquisition, Validation, Methodology; Elad Noor, Conceptualization, Data curation, Software, Formal analysis, Supervision, Funding acquisition, Validation, Visualization, Methodology, Writing – original draft, Project administration, Writing – review and editing; Ron Milo, Conceptualization, Resources, Formal analysis, Supervision, Funding acquisition, Validation, Visualization, Methodology, Writing – original draft, Project administration, Writing – review and editing

## Author ORCIDs

Eliya Milshtein ⓘ http://orcid.org/0009-0005-3405-1054
Elad Noor ⓘ http://orcid.org/0000-0001-8776-4799
Ron Milo ⓘ https://orcid.org/0000-0003-1641-2299

Joint Public Review: https://doi.org/10.7554/eLife.88793.4.sa1
Author Response https://doi.org/10.7554/eLife.88793.4.sa2

---

# Additional files

## Supplementary files

- Supplementary file 1. iterative evolution mutations.
- Supplementary file 2. Compact and evolved labeling run curated peaks summary.
- Supplementary file 3. compact vs evolved growth experiment DATA 45 mM 5 pctO2.
- Supplementary file 4. compact vs evolved growth experiment MAP 45mM 5 pctO2.
- Supplementary file 5. pgi kinetic assays data.
- Supplementary file 6. G6PvsF6PResults.
- Supplementary file 7. PPP metabolites.
- Supplementary file 8. Growth experiment WT vs DM FDH variants $CO_2$ 40mMFor DATA.
- Supplementary file 9. Growth experiment WT vs DM FDH variants $CO_2$ 40mMFor MAP.
- Supplementary file 10. Growth experiment WT vs DM FDH variants $CO_2$ NoFormate DATA.
- Supplementary file 11. Growth experiment WT vs DM FDH variants $CO_2$ NoFormate MAP.
- Supplementary file 12. rpoB crp metabolomics.
- Supplementary file 13. proteomics Evo Anc.

- Supplementary file 14. proteomics WT variants.
- Supplementary file 15. Compact AutoColi dasgib log.
- Supplementary file 16. Scaffoldless Autocoli dasgib log.
- Supplementary file 17. Primer list.
- MDAR checklist

## Data availability

All data generated or analyzed during this study are included in the manuscript and supporting files.

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
