## [Editor Report · eLife assessment]

This is an **important** follow-up study to a previous paper in which the authors reconstituted CO_2_ metabolism (autotrophy) in *Escherichia coli*. Here, the authors define a set of just three mutations that promote autotrophy, highlighting the malleability of *E. coli* metabolism. The authors make a **convincing** case that mutations in pgi are loss-of-function mutations that prevent metabolic efflux from the reductive pentose phosphate autocatalytic cycle, and their data suggest possible roles of mutations in two other genes - *crp* and *rpoB*. This research will be particularly interesting to synthetic biologists, systems biologists, and metabolic engineers aiming to develop synthetic autotrophic microorganisms.

---

## [Referee Report · Joint Public Review]

The authors previously showed that expressing formate dehydrogenase, rubisco, carbonic anhydrase, and phosphoribulokinase in *Escherichia coli*, followed by experimental evolution, led to the generation of strains that can metabolise CO_2_. Using two rounds of experimental evolution, the authors identify mutations in three genes - pgi, rpoB, and crp - that allow cells to metabolise CO_2_ in their engineered strain background. The authors make a strong case that mutations in pgi are loss-of-function mutations that prevent metabolic efflux from the reductive pentose phosphate autocatalytic cycle. The authors also use proteomic analysis to probe the role of the mutations in crp and rpoB. While they do not reach strong conclusions about how these mutations promote autotrophic growth, they provide some clues, leading to valuable speculation.

Comments on revised version:

The authors have thoroughly addressed the reviewers' comments. The major addition to the paper is the proteomic analysis of single and double mutants of crp and rpoB. These new data provide clues as to the role of the crp and rpoB mutations in promoting autotrophic growth, which the authors discuss. The authors acknowledge that it will require additional experiments to determine whether the speculated mechanisms are correct. Nonetheless, the new data provide valuable new insight into the role of the crp and rpoB mutations. The authors have also expanded their description of the crp and rpoB mutations, making it clearer that the effects of these mutations are likely to be distinct, albeit with potential for overlap in function.

---

## [Author Response]

The following is the authors’ response to the previous reviews.

**Recommendations for the authors:**
The single-mutant and double-mutant crp/rpoB strains were made by co-transduction with a nearby gene deletion (kanR-marked). I couldn't tell from the methods section whether these mutants, e.g., crp-H22N delta-chiA, were compared to wild-type cells or deletion mutants, e.g., delta chiA, in the proteomics experiments. I encourage the authors to explain this more clearly in the methods section, and to briefly mention in the Results section and relevant figure legends that the crp/rpoB mutant strains (and possibly the "wild-type" strains) also have gene deletions. If the comparison "wild-type" strains are fully wild-type (i.e., not deleted for chiA/yjaH), it is especially important to mention this in the Results section and the figure legends since the phenotypic changes could be due to the gene deletions rather than the mutations in crp/rpoB

We appreciate and agree with the editor's suggestion to clarify this point.

Accordingly, we have made the following changes to the text:

p11 L30-34 in the main text:

"The second experiment similarly compared an engineered BW25113 (BW) strain, containing the two regulatory mutations from the compact set (i.e., crp H22N and rpoB A1245V) together with the deletions used to insert them (see methods and DataS1 file), to a “wild type” BW strain (a corresponding knockout strain without the mutations, see methods)."

p28 under Chemostat proteomics experiment L13-16 in methods:

"The starting volume of each bioreactor was 150 ml M9 media supplemented with either30 mM and 10mM D-xylose for the evolved and ancestor samples or only 10mMD-xylose for BW including compact set mutations and/or the deletions used for their insertions (DataS1 file). The minimal media also included trace elements and vitamin B1 was omitted."